# Convolutional Neural Network Approach Based on Multimodal Biometric System with Fusion of Face and Finger Vein Features

**DOI:** 10.3390/s22166039

**Published:** 2022-08-12

**Authors:** Yang Wang, Dekai Shi, Weibin Zhou

**Affiliations:** School of Electronic Information and Automation, Tianjin University of Science and Technology, Tianjin 300453, China

**Keywords:** CNN, dual-channel biometric identification system, biometric fusion, face recognition, finger vein recognition, identification system

## Abstract

In today’s information age, how to accurately identify a person’s identity and protect information security has become a hot topic of people from all walks of life. At present, a more convenient and secure solution to identity identification is undoubtedly biometric identification, but a single biometric identification cannot support increasingly complex and diversified authentication scenarios. Using multimodal biometric technology can improve the accuracy and safety of identification. This paper proposes a biometric method based on finger vein and face bimodal feature layer fusion, which uses a convolutional neural network (CNN), and the fusion occurs in the feature layer. The self-attention mechanism is used to obtain the weights of the two biometrics, and combined with the RESNET residual structure, the self-attention weight feature is cascaded with the bimodal fusion feature channel Concat. To prove the high efficiency of bimodal feature layer fusion, AlexNet and VGG-19 network models were selected in the experimental part for extracting finger vein and face image features as inputs to the feature fusion module. The extensive experiments show that the recognition accuracy of both models exceeds 98.4%, demonstrating the high efficiency of the bimodal feature fusion.

## 1. Introduction

In recent years, with the increasing level of science and technology, people direct progressively more attention to information security [1]. Traditional authentication methods, such as using account numbers and passwords are easily replaced by impersonation once they are stolen. Biometrics is an emerging security technology and one of the most promising technologies in this century. Compared with traditional identification, biometrics has many advantages, such as the technology will not be lost, stolen or copied. Biometrics mainly refers to a technology that authenticates identity through measurable physical or behavioral biometrics [2]. Biological features are divided into two categories: Physical features and behavioral features [3]. Physical features, such as fingerprints, retinas, and faces, are mostly inborn, and behavioral features, such as gait and keystrokes, are mostly acquired due to habit [4].

Primarily, a biometric identification system involves four processes: Image acquisition, image processing feature extraction, feature value comparison, and individual identification. There are two biometric systems, single-modal and multimodal. A single-modal biometric system refers to the use of a single biometric feature to identify users. Although the current biometric technologies, such as face recognition [1], fingerprint recognition [5], and iris recognition [6] are relatively mature, with the outbreak of COVID-19, single biometric technologies, such as fingerprint recognition have suffered serious challenges. Face recognition hidden behind masks fails, and fingerprint recognition with protective gloves cannot be unlocked. Single biometric technologies represented by fingerprints and faces are increasingly difficult in application.

With the upgrading and development of biometric identification technology, biometric identification is constantly changing from single to multimodal. Through careful design and fusion algorithm, it can realize the combination of face, fingerprint, fingerprint vein, iris, voice print, and other biometrics, which can lead to complementary information and further improvement of recognition accuracy. Multimodal biometrics refer to the integration or fusion of multiple human biometrics, using the respective advantages of biometrics and combining different feature fusion algorithms to make the identification process more accurate and secure [7]. Compared with single biometric identification technology, multimodal biometric identification technology has the advantages of high identification accuracy, higher security, and wider application range.

Although the current single-modal biometric technology has been relatively mature, this single-mode biometric recognition technology is not only affected by the external environment in practical applications, but also affected by the limitations of single-mode biometrics itself, which greatly limits its application scenarios and reduces the accuracy of identity recognition. For example, in fingerprint recognition, fingerprint damage and finger wetness will affect the correct rate of collection and recognition. In face recognition, the growth of age and wearing masks will affect its recognition accuracy. To overcome these problems, we propose a bimodal recognition method that fuses face and finger veins. Compared with the three-modal, four-mode, and other modes, the finger vein and face bimodal not only reduce the complexity of time, algorithm, design, and production, but also complement the advantages of in vivo biometrics (finger veins) and in vitro biometrics (faces). It breaks the application limitations of single biometrics and improves the security and accuracy of identity information.

In this study, we chose face and finger veins as the targets for the following reasons: Faces are chosen since facial features are the most pronounced, and more data are used, making them more precise. Face recognition does not require any contact, it can be identified by air. Finger veins are chosen since they are internal physiological features that are difficult to forge and have a high level of safety [2]. It has the advantages of enabling non-contact measurement, good hygiene, and easy acceptance for users. In addition, the use of each person’s finger vein is unique and cannot be forged, especially for places with high safety requirements.

Traditional biometric systems primarily have four components: Pre-processing, feature extraction, matching, and decision-making phases [8]. Feature extraction methods can affect the system significantly. In view of the excellent performance of convolutional neural networks in image recognition and image feature extraction tasks, this study aims to deeply study the application of convolutional neural network algorithms in finger vein and face biometric recognition. In this paper, a bimodal biometric system based on a deep learning model of finger veins and face images is proposed. First, the finger vein and face images were acquired, and the region-of-interest (ROI) was intercepted on the finger vein images. For the problem of the lack of data in the finger veins, the image enhancement method was adopted, and then the finger vein and face images were input into the dual-channel convolutional neural network to extract the features. Feature fusion is performed before the fully connected layer, and weights are assigned according to the confidence level of each feature, and finally user identification is performed.

The rest of the paper is organized as follows: Section 2 provides a brief description of the relevant research. Section 3 proposes the method used in this article. Section 4 discusses and analyzes the results. Finally, Section 5 concludes the paper and discusses potential future work.

## 2. Related Work

Multimodal biometric technology refers to the fusion of more than two kinds of single biometric features as a new feature for identification [9]. According to the location of the fusion, it can be divided into image layer fusion, feature layer fusion, matching layer fusion, and decision layer fusion [10].

Haghiatt et al. [11] proposed discriminative correlation analysis (DCA), which performs an effective feature fusion by maximizing the pairwise correlations across the two feature sets and, at the same time, eliminating the between-class correlations and restricting the correlations to be within the classes. Shaheed et al. [12] considered the issues of convolutional neural networks (CNNs). The authors presented a pre-trained CNN network named Xception model based on depth-wise separable CNNs with residual connection, which is considered as a more effective, less complex neural network to extract robust features. Their proposed method for the SDUMLA database achieved an accuracy of 99% with an F1-score of 98%. While on THU-FVFDT2, the proposed method obtained an accuracy of 90% with an F1-score of 88%. Qi et al. [13] proposed a biometric method based on three biometric patterns of face, iris, and palm print, and introduced a strategy of fusion of multiple features. Cherrat et al. [14] proposed a hybrid system based on a multi-biometric fingerprint, finger vein, and face recognition system that combines a convolutional neural network (CNN), Softmax, and a random forest (RF) classifier, using K-means and density-based spatial clustering of application with Noise (DBSCAN) algorithms that employ image preprocessing to separate foreground and background areas. Experiments have shown that multimodality can provide accurate and efficient matching, which can be significantly better than the single-modal representation accuracy.

Few studies focus on identifying users through behavioral biometrics, and in the process of research, feature recognition and feature extraction are more difficult since user behavior is non-repeatable and variable. Abinaya et al. [15] used two different unmoded biometric keystrokes (typography) and acoustics (speech) to identify the use of pre-trained deep learning models, using weighted linear feature-level fusion, and trained by CNN classifier simulation. The results show that the system has good accuracy. Ding et al. [16] proposed a deep learning structure consisting of a neural network (CNN) and a three-layer stacked autoencoder, and then concatenated the extracted features to form a high-dimensional feature vector. This method achieves a recognition rate of up to 99.0% on the Labeled Faces in the Wild (LFW) database. Chawla et al. [17] investigated the finger vein recognition problem. They employed one-shot learning model namely the Triplet loss network model and evaluated its performance. The extensive set of experiments that they have conducted yield classification and correct identification accuracies in ranges upwards of 95% and equal error rates less than 4%.

In recent years, research on finger vein recognition has been particularly active. For example, Alay et al. [18] combined three CNN models of iris, finger vein, and face to fuse at the feature and fractional levels, respectively, using the VGG-16 model, where the accuracy of feature-level fusion was 100% and the accuracy of different score-level fusion methods was 100%. Ammour et al. [8] proposed a new feature extraction technique for a multimodal biometric system using face–iris traits. The iris feature extraction is carried out using an efficient multi-resolution 2D Log-Gabor filter to capture textural information in different scales and orientations. On the other hand, the facial features are computed using the powerful method of singular spectrum analysis (SSA) in conjunction with the wavelet transform. Kim et al. [19] studied a biometric system capable of recognizing finger veins and finger shapes, and constructed a CNN biometric system, using fusion methods, such as weighted sum and Bayesian rules to fuse two features on the fractional level.

Yang et al. [20] proposed a research method for the fusion of finger veins and fingerprints, developed a feature-level fusion strategy for three fusion options, and combined the details of finger vein fingerprint image features. Soleymani et al. [21] proposed a fusion method for the fingerprints of the human face and the iris. Multiple features are extracted from each modal-specific CNN in multiple different convolutional layers for joint feature fusion, optimization, and classification. Waisy et al. [22] proposed a deep learning method based on the left and right iris images of people, using a hierarchical fusion method to fuse the results, which is called IrisConvNet. The recognition rate of the database used is 100% and the recognition time is less than 1 s. Ren et al. [23] proposed a new finger vein image encryption scheme, which applies Rivest–Shamir–Adleman encryption technology to finger vein image encryption. In addition, a complete cancelable finger vein recognition system with template protection is proposed to ensure the security of the user’s vein template while maintaining the recognition performance. Jomaa et al. [4] aimed at the serious vulnerability of fingerprint system in presentation attack (PA), they proposed a novel architecture based on end-to-end deep learning to fuse between neuroprinting and ECG, improving PA detection in fingerprint biometrics and using EfficientNet to generate fingerprint features. Experimental results show that the architecture produces better average classification accuracy than the single fingerprint. Tyagi et al. [24] proposed highly accurate and robust multimodal biometric identification as well as recognition systems based on fusion of face and finger vein modalities. The feature extractions for both face and finger vein are carried out by exploiting deep convolutional neural networks.

In view of the selected biological features that are easy to be affected by external factors and different fusion locations, this study selected face and finger vein, complementing the advantages of internal biological features (finger vein) and external biological features (face), and integrating in the feature layer with the most rich and effective information. Not only does it make up for the shortcomings of single biometric recognition and improve the accuracy of identity recognition, but also compared with the three-mode, four-mode, and other modes, the finger vein and face bimodal reduces the time, algorithm, design, and production complexity.

## 3. Method

In this study, a two-channel CNN feature fusion framework is proposed as shown in Figure 1, and the fusion occurs at the feature layer. The framework is mainly divided into three parts, namely the feature extraction part, the feature fusion part, and the classification recognition part. In the feature extraction, the collected finger vein pictures and face pictures are mainly pre-processed, such as the interception of the finger vein area of interest and data enhancement, and then the biometrics are convoluted into the neural network model, and the biometrics of the images are extracted through multi-layer convolution and pooling layers. In the feature fusion module, fusion Conv is first performed to reduce dimensions and then pass through the Softmax layer, which obtains the self-attention weights and multiplies the features obtained by feature extraction, and then channel Concat cascades the two features together to obtain the fusion features of finger vein and face (Fv + Face_feature). To prevent the loss of some feature information during feature fusion, the finger vein features and face features after feature extraction are obtained, as well as feature cascading after the feature three-channel Concat. The classification recognition part is mainly classified in the fully connected layer.

### 3.1. Network

Mode convolutional neural network is a feedforward neural network, which is the mainstream technology in image processing today [24]. Using CNN for processing pictures can not only effectively reduce the dimension of large data volume to small data amount, but also can effectively retain the characteristics of pictures, which complies with the principle of image processing. Similar to other neural networks, CNN networks also contain several parts: Input layer, hidden layer, and output layer. The convolutional layer [25] is the core layer of building a convolutional neural network, the convolutional layer is composed of multiple convolutional units, and the parameters of each convolutional unit are optimized by backpropagation algorithms. Convolutional operations are mainly to extract the features of the image, the first few layers are mainly to extract the low-level features of the image, such as color and other information, with the increase in the convolutional layer, the multi-layer network can extract more complex image features. Linear rectification mainly refers to the ReLU function of the activation function operation, which can realize the non-linear mapping function of the network, increase the expression ability of the network, and have a good effect on feature extraction. The input of the pooling layer is multiple feature mapping and pooling input operation. After convolution, the dimensional features of the image still divide the feature matrix into several single blocks to take its maximum or average, which can play a role in dimensionality reduction and reduce the calculation of the network and avoid overfitting. Overfitting is also known as over-fitting. Due to over-fitting of the training samples, the ability to accept samples other than the training samples is poor, and the model cannot have good generalization ability. The input of the pooling layer is multiple feature mapping and pooling the input. After convolution, the dimensional features of the image are still many, and the feature matrix is divided into several single blocks to take their maximum or average values, which can play a role in reducing dimensionality, reducing network computation, and avoiding overfitting. As the last layer of the CNN model, the fully connected layer combines all local features and the feature matrix of each channel into vector representations, and calculates the score of each final class, as shown in Figure 2.

To demonstrate the high efficiency of bimodal feature fusion, the feature extraction networks of two representative AlexNet [25] and VGG-19 [26] networks are selected as the benchmark [27,28], discarding their fully connected layers and using only the convolutional and pooling layers before the fully connected layers.

AlexNet: The network, designed by 2012 ImageNet competition winner Hinton and his student Alex Krizhevsky, consists of five convolutional layers (conv) and three fully connected layers (fc) as shown in Figure 3. The activation function uses ReLU, and the entire network has more than 62 million trainable parameters. The input is a 224 × 224 × 3 image, and the output is a 1000-dimensional vector corresponding to the probability of each classification. The first, second, and fifth convolution layers add a pooling layer (MaxPool) with a kernel of 3 × 3 and a stride of 2, which can improve the accuracy of the pooling layer, as shown in Figure 3.

VGG-19: The network has 19 layers with 16 convolutional layers and 3 fully connected layers as shown in Figure 4. The convolution kernels are all 3 × 3 in size, which reduces the network parameters by repeatedly stacking the 3 × 3 convolution kernels rather than a large convolution kernel. Compared with the AlexNet network, VGG-19 is able to extract deeper features of images. Therefore, in the feature extraction module, the first 16 convolutional layers of VGG-19 are used to extract finger vein face image features.

### 3.2. Data Preprocessing

Face images use the face public dataset CASIA-WebFace [29], which performs pre-processing, such as resize and normalization in the network. Since the final experimental data worked well, no other complex pre-processing operations were performed.

For finger veins, since the SDUMLA-FV dataset [30] does not provide the region-of-interest (ROI) of the finger vein, an interception of the ROI region of the dataset is required to remove excessive background information which is likely to be useless. First, the Prewitt edge detection operator is used to detect the upper and lower edges in the vertical direction of the original diagram of the finger vein, and for the phenomenon of pseudo-edges, the pseudo-edges are removed by setting the connection domain threshold. Use least squares linear regression to fit the central axis of the finger, and correct the image rotation according to the angle between the fitted line and the horizontal line. Fit the inner tangent of the upper and lower edges of the finger. According to the brightness change trend in the horizontal direction of the image, select the knuckle (brightness peak). Finally, the venous ROI area of the finger is intercepted.

To obtain clear finger vein lines, contrast-limited adaptive histogram equalization (CLAHE) is also required for the captured region-of-interest (ROI) images, and a Gabor filter is added after the CLAHE image enhancement to remove noise. The ROI original image, after CLAHE image enhancement and Gabor filtering, can get clear vein lines compared with the original image. The finger vein dataset provided only 6 images of veins per finger, and to prevent overfitting during CNN model training, we amplified the data for each type of finger vein, including random translation, rotation, cropping, brightness adjustment, and contrast adjustment of the image, expanding the original 6 images per class to 36 pictures. The FV-USM dataset [31] provides ROI images, thus only image enhancement and augmentation are required for this dataset, as shown in Figure 5.

### 3.3. Feature Layer Fusion

In the feature layer fusion method, the face image feature (Face_feature) and the finger vein image feature (Fv_feature) are taken as the input to the self-attention mechanism to obtain their respective attention weights. The vein and face features are combined with their respective attention weights, and they are cascaded to form cascade features (Fv + Face_feature). The cascade features (Fv + Face_feature) are taken as the input of the fusion module to convolute the cascade features to extract the deeper features and obtain the fusion features (Fusion feature). To prevent the loss of some feature information during feature fusion and ensure the maximization of effective feature information, the fusion features are fused again with the single finger venous features and the single face features into new features and output in Figure 6.

The self-attention mechanism proposed in this article consists of three convolutional layers and one Softmax layer, and the structure of the convolutional layer is shown in Table 1. The convolutional kernel size of the first convolution is 1 × 1, padding is 0, and the convolutional kernel size of the second and third convolutions is 3 × 3, and padding is 1. After the convolutional layer, the ReLU activation function is used, and the ReLU activation function can not only quickly converge the network, but also solve the problem of gradient disappearance. Table 1 shows the convolutional structure in the self-attention mechanism network, and Table 2 is the convolutional structure in the Fusion Block fusion module proposed in this paper.

The ReLU function formula is as follows:ReLU(x) = max(0,x)(1)

The input of the whole dual-channel convolutional neural network is two images (one finger vein image and one face image), and new features are formed after the feature extraction module and feature fusion module. New features were used as input to the classification identification module.

## 4. Experimental Results and Analysis

### 4.1. Experimental Environment and Data Distribution

The language used in this experiment is Python3.8, the software environment is Ubuntu18.04 (64-bit), and the framework used for deep learning is Pytorch 1.7.1, CUDA11.0, cuDNN8.0, using two NVIDIA GeForce RTX 3080Ti GPUs.

To demonstrate the effectiveness of the method studied in this paper, we used three publicly available datasets, CASIA-WebFace [29], Finger Vein USM (FV-USM) [31], and SDUMLA-FV [30], to test the proposed bimodal feature fusion algorithm and compare it with single-modal biometric identification. The SDUMLA-FV dataset was created by Shandong University and captured a total of 3816 images. A total of 636 classes contained 106 finger vein images of the left and right index fingers, 6 middle fingers, and ring fingers. Each image is stored in “BMP” format (320 × 240) pixel size. FV-USM is a Malaysian Polytechnic finger vein dataset with 2952 maps. A total of 492 types of fingers contains 6 middle finger images and 123 images of the left and right hands, and the advantage of using this dataset is that it provides already intercepted region-of-interest (ROI) images. The CASIA-WebFace dataset is one of the most widely available datasets for applied face recognition, which collects face images on the network with a total of 10,575 classes and 494,414 images.

This experiment used two finger vein public datasets, FV-USM, SDUMLA-FV, and one face dataset, CASIA-WebFace. In single-mode experiments, each sample was amplified to 36 pictures using the data amplification method, and 636 categories were randomly selected for the face dataset. Additionally, each dataset divides the training, validation, and testing. Sixty percent of patterns are assigned to the training set, 10% to the validation set, and 30% to the test set. To increase the generalization capacity of the network, n × n is used to match each finger vein image belonging to the same class with each face image. For the dataset of 636 classes, the training set total is 636 × 21 × 21 = 280,476 images, validation set total is 636 × 3 × 3 = 5724 images, and test set total is 636 × 12 × 12 = 91,584 images. Then, the total amounts were divided into training sets, validation sets, and test sets.

### 4.2. Testing and Analysis of Biometric Systems

#### 4.2.1. Performance Evaluation

This article mainly provides a bimodal fusion method. To evaluate the structural performance of this model, we measure the performance of the model by the following indicators.

Confusion Matrix: In the field of image recognition, it is mainly used to compare the relationship between the classification results and the actual predicted values. Table 3 shows the confusion matrix, positive represents positive samples, and negative represents negative samples. Each column represents the predicted value, and each row represents the actual value.

*Accuracy rate*: In all the samples, the proportion of the correct number of samples in the total number of samples is predicted; the mathematical expression is as follows:
(2)$$accuracy_{\_}rate=\frac{TP+TN}{\begin{array}{c} TP+TN+FP+FN \end{array}}$$

*Precision*: In all the samples with positive class prediction result, the proportion of the samples is actually positive class and the prediction result is also positive class number; the mathematical expression is as follows:(3)$$precision=\frac{TP}{\begin{array}{c} TP+FN \end{array}}$$

*Recall rate*: In all the samples that are actually positive, the proportion of the number of samples is predicted as positive; the mathematical expression is as follows:(4)$$\;recall=\frac{TP}{\begin{array}{c} TP+FN \end{array}}$$

ROC curve: The abscissa of ROC curve is negative positive rate (false positive rate, FPR). The ordinate is the true class rate (true positive rate, TPR).
(5)$$FPR=\frac{FP}{\begin{array}{c} FP+TN \end{array}}$$
(6)$$TPR=\frac{TP}{\begin{array}{c} TP+FN \end{array}}$$

The area under the ROC curve represents AUC (area under curve), and its value can directly reflect the performance of the classification model. The value range is between 0.1 and 1. The closer the value is to 1, the better the performance of the classification model.

P-R curve: The P-R curve is constructed with precision as the ordinate and recall as the abscissa, and the area enclosed by the coordinate axis indicates the average accuracy (average precision, AP). The value of AP reflects the performance of the classification model. The closer the AP value is to 1, then the classification model has good performance.

#### 4.2.2. Experimental Results and Analysis of Single-Mode Identification

In the single-mode biometric experiment, three datasets, SDUMLA-FV, FV-USM, and CASIA-WebFace, were used to experiment on the CNN network framework (AlexNet, VGG-19). As can be seen from Table 4, in the two network models, the identification accuracy of single-modal experiment can reach 87.57% and the lowest is 45.61%. Compared with Yuancheng’s face recognition experiment [15], the accuracy of single-modal identification is not high and the number of parameters is large.

#### 4.2.3. Experimental Results and Analysis of Multimodal Recognition

In Table 5, the accuracy of the recognition of face and finger veins in the feature layer fused under the two CNN network models is shown. Compared with the single-mode experimental results, the accuracy rate of AlexNet as a feature extraction network was above 99.3%. In the experiment using VGG-19 as the feature extraction network, the feature fusion experimental recognition accuracy of the SDUMLA-FV dataset and the CASIA-WebFace face dataset reached 99.98%, and the feature fusion experimental recognition accuracy of the USM-FV dataset and the CASIA-WebFace face dataset reached 98.42%. Through the experimental data, we can prove that the proposed method can greatly improve the accuracy and effectively shorten its model parameters.

The following figure is the plotted ROC curve. There are two methods of ROC curve drawing, which are micro average and macro average. Micro average refers to calculating the prediction accuracy of each sample model, establishing a global confusion matrix, and then calculating the corresponding indicators globally to draw the micro average ROC curve of the dataset, regardless of the category of each test sample in the dataset. Macro-averaging refers to the separation of each class of the dataset, then calculates the accuracy of each class, establishing the confusion matrix, and finally averaging the average of the ROC curve and the AUC values.

Figure 7 shows the ROC curves of the FV-USM dataset and the SDUMLA-FV dataset combined with the CASIA-WebFace face dataset to use the bimodal feature fusion method on the AlexNet feature extraction network. As you can see from the figure, the AUC of the micro average and the macro average is both 1.

Figure 8 shows the ROC curves of experimental results for different datasets using the bimodal feature fusion method on the VGG-19 feature extraction network. As you can see from the figure, the AUC at both the micro average and the macro average is 1.

Table 6 shows a bimodal dataset consisting of two different finger vein datasets and a face dataset, and the area AUC of the ROC curve and the curve surrounded by the coordinate axis is summarized on three different feature extraction networks. As you can see from the summary table, the AUC is all 1.

In summary, according to Yasen’s experimental analysis [25], the closer the ROC curve is to the upper left corner, the better the performance. Experimental data show that the bimodal feature fusion method proposed in this paper has good classification performance in different bimodal datasets and different feature extraction networks.

To evaluate the proposed bimodal feature fusion recognition method more objectively and accurately, we plotted the P-R curves for each experimental result. Figure 9 shows the P-R curve results of different datasets using the bimodal feature fusion method on the AlexNet feature extraction network. Among them, the bimodal feature fusion experiment of the FV-USM dataset has an area AP of 0.88 surrounded by coordinate values, and the area AP of the SDUMLA-FV dataset surrounded by coordinate axes is 0.9.

Figure 10 shows the P-R curve results of different datasets using the bimodal feature fusion method on the VGG-19 feature extraction network. As can be seen from the figure, the area AP enclosed by the P-R curve and the coordinate values is 0.92 and 0.8, respectively.

Table 7 shows a summary of the values of two different bimodal datasets on the P-R curve as well as the coordinate axis surrounded by the P-R curve and coordinate axis on two different feature extraction networks.

For the P-R curve index, the average accuracy AP value is greater than 0.88 and the highest is 0.92 in all experimental results. Compared with Yang’s experiments [31], they used the precision-recall (P-R) curve to measure the accuracy-recall trade-off due to unbalanced datasets and reported sensitivity and specificity on the receiver operating characteristic curve. The area under the precise-recall curve showed an average accuracy of 0.874, indicating that the proposed method has good classification performance.

## 5. Conclusions

This study proposes a two-modal feature layer fusion method based on convolutional neural network, combining the advantages of in vivo biometric features (finger vein) and in vitro biometric features (human face). This method introduces the weight of the self-attention mechanism to update the different features within the feature fusion module, and adopts the residual structure to maximize the effective feature information. The feature extraction was performed on the AlexNet and VGG-19 feature extraction modules, respectively, and then the bimodal feature fusion method proposed in this paper was used. In multimodal experiments, the lowest identification accuracy of the experimental results was 98.84%, and the highest identification accuracy was 99.98%. In addition, the ROC curve and P-R curve of the fusion experiment are plotted, which verifies that the proposed method has very good classification performance. The research in this paper is currently limited to computers with high computing power of GPUs, and cannot be deployed to mobile embedded terminals, which greatly limits the application in practice. Subsequently, the network results of bimodal fusion will be lightweight, and the fusion model will be ported to an embedded terminal within an acceptable range.

## Figures and Tables

**Figure 1 sensors-22-06039-f001:**
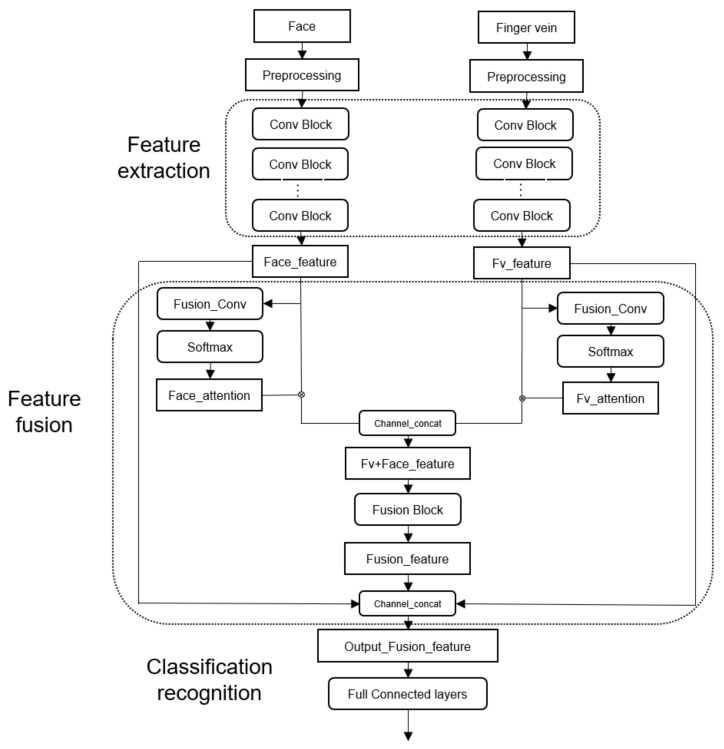
Bimodal feature fusion framework.

**Figure 2 sensors-22-06039-f002:**
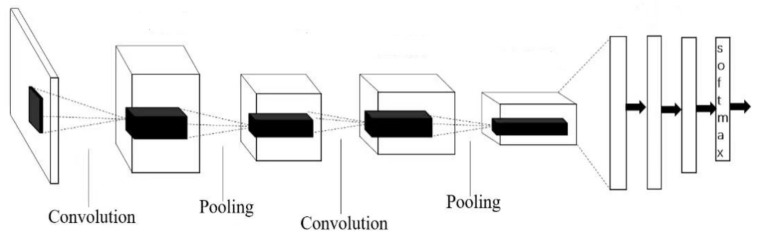
Schematic diagram of convolutional neural network structure.

**Figure 3 sensors-22-06039-f003:**
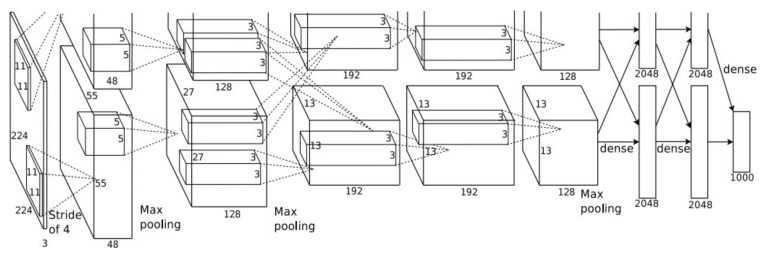
Schematic diagram of the AlexNet model architecture.

**Figure 4 sensors-22-06039-f004:**
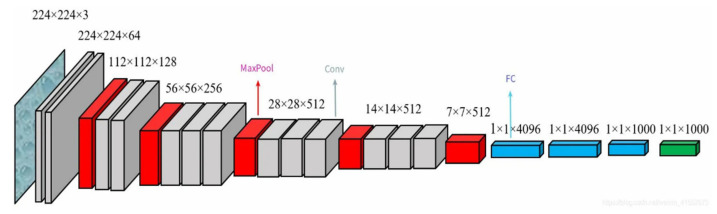
Schematic diagram of the VGGNet model architecture.

**Figure 5 sensors-22-06039-f005:**
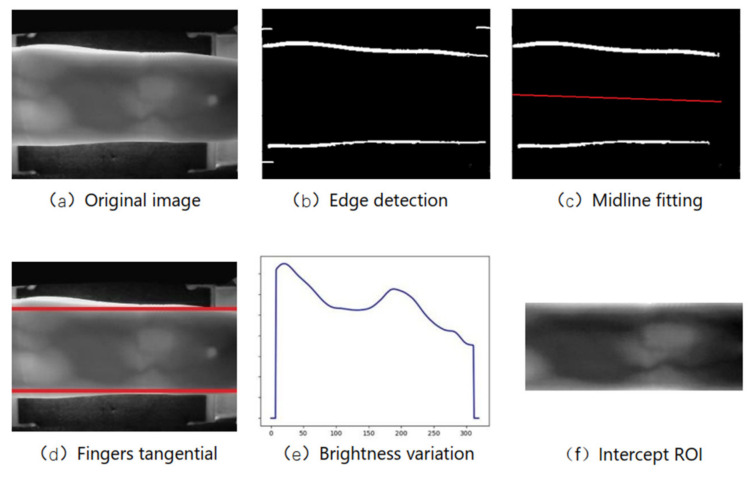
SDUMLA-FV dataset image preprocessing.

**Figure 6 sensors-22-06039-f006:**
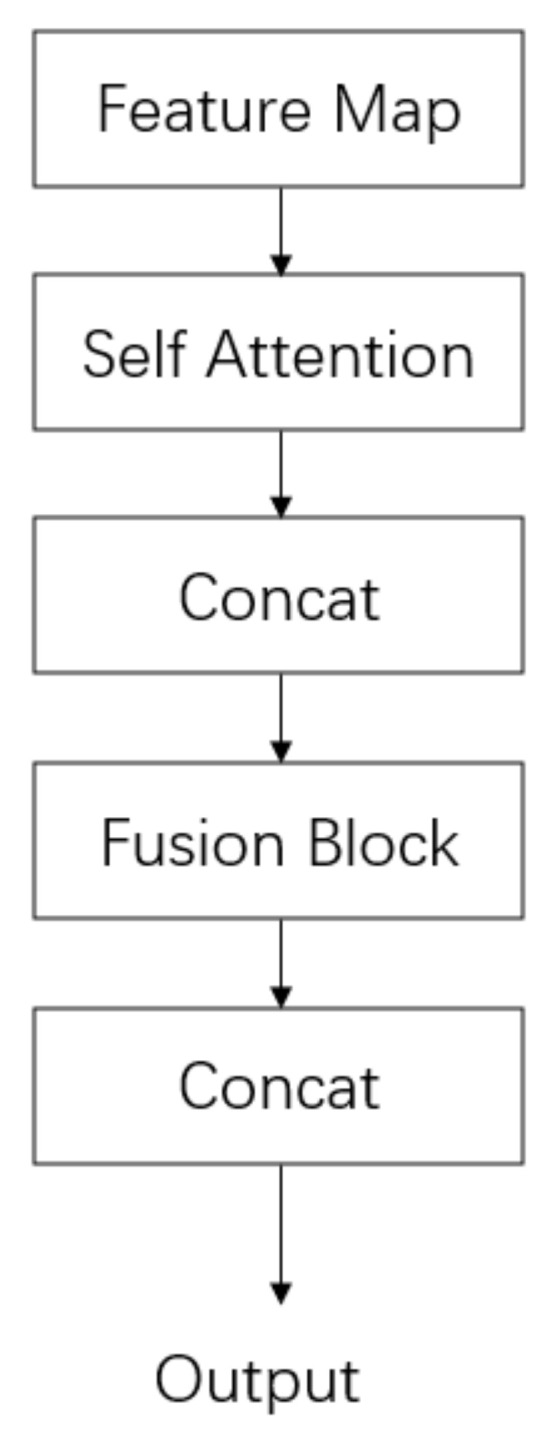
Schematic diagram of the feature fusion module framework.

**Figure 7 sensors-22-06039-f007:**
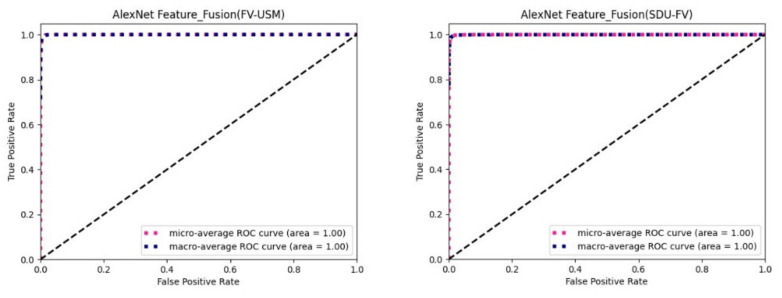
Fusion of identified ROC curves on AlexNet feature extraction network.

**Figure 8 sensors-22-06039-f008:**
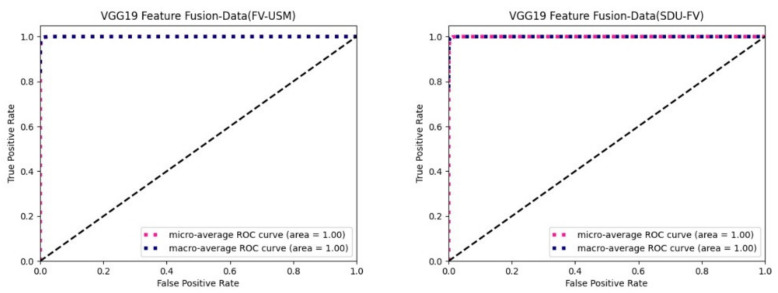
Fusion of identified ROC curves on the VGG-19 feature extraction network.

**Figure 9 sensors-22-06039-f009:**
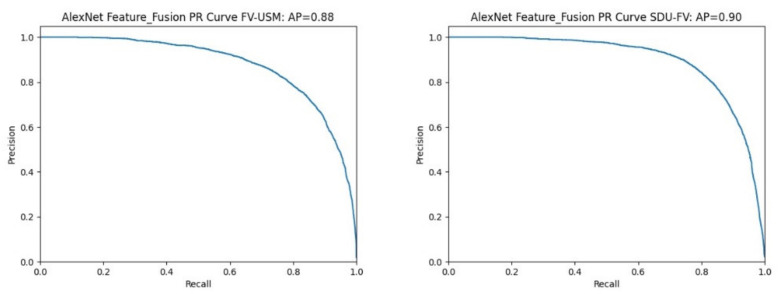
Fusion of identified P-R curves on the AlexNet feature extraction network.

**Figure 10 sensors-22-06039-f010:**
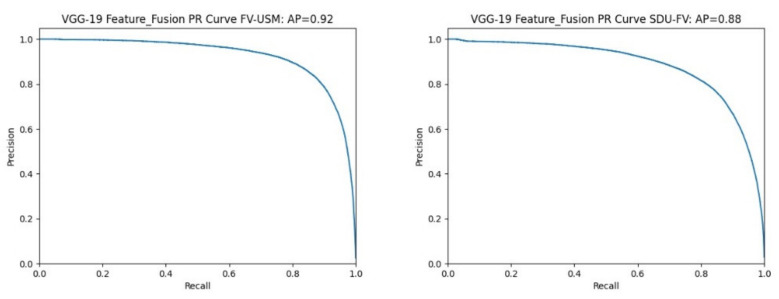
Fusion of identified P-R curves on the VGG-19 feature extraction network.

**Table 1 sensors-22-06039-t001:** Convolutional architecture in self-attention networks.

Layer Name	In Channel	Out Channels	Convolutional Kernel Size	Padding	Stride
Conv1	a	a/2	1	0	1
Conv2	a/2	a/4	3	1	1
Conv3	a/4	1	3	1	1

**Table 2 sensors-22-06039-t002:** Convolutional layer structure in fusion block.

Layer Name	In Channel	Out Channels	Convolutional Kernel Size	Padding	Stride
Conv1	2 × a	a	1	0	1
Conv2	a	a/2	3	1	1
Conv3	a/2	a	3	1	1

**Table 3 sensors-22-06039-t003:** Confusion matrix.

	Predicted Value	Positive	Negative
Actual Value			
Positive		True Positive (*TP*)	False Negative (*FN*)
Negative		False Positive (*FP*)	True Negative (*TN*)

**Table 4 sensors-22-06039-t004:** Results of a single-mode experiment.

	Dataset	Parameter Quantity	Test Set Accuracy		
Model			SDUMLA-FV	USM-FV	CASIA-WebFace
AlexNet		16,630,440	0.7020	0.4561	0.5395
VGG-19		143,667,240	0.8757	0.6734	0.5575

**Table 5 sensors-22-06039-t005:** Results of multimodal experiments.

	Dataset	Parameter Quantity	Test Set Accuracy	
Model			SDUMLA-FV + CASIA-WebFace	USM-FV + CASIA-WebFace
AlexNet-Fusion		9,858,994	0.9990	0.9935
VGG-19-Fusion		45,229,938	0.9998	0.9842

**Table 6 sensors-22-06039-t006:** AUC summary of different datasets under different feature extraction networks.

	Feature Extraction Network	AUC	
Dataset		AlexNet	VGG-19
FV-USM + CASIA-WebFace		1	1
SDUMLA-FV + CASIA-WebFace		1	1

**Table 7 sensors-22-06039-t007:** Summary of AP values of different datasets under different feature extraction networks.

	Feature Extraction Network	AP	
Dataset		AlexNet	VGG-19
FV-USM + CASIA-WebFace		0.88	0.92
SDUMLA-FV + CASIA-WebFace		0.90	0.88

## Data Availability

Not applicable.

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
