# Peer review of "Convolutional Neural Network Approach Based on Multimodal Biometric System with Fusion of Face and Finger Vein Features"

_sensors, 2022, doi:10.3390/s22166039_

Round 1

Reviewer 1 Report

This study provides a recognition method based on the merging of facial and finger vein data to address the issue of using a single biometric for complicated and diverse authentication circumstances. The problem is interesting. There are some good results but there are some issues which need to be fixed too. 

- In the abstract, you need to show not only the accuracy measure to prove the efficiency of the proposed model but also the FAR and FRR which are very important in face recognition systems. 

- Problem gap: the problem gap is not clear or not well-identified. There are many articles about multimodal biometric identification systems, see examples below. Before talking about what you will do, in the introduction, line 58, you need to show what are the limitations of previous work, namely face and finger vein fusion as this is your choice

Ammour, B., Boubchir, L., Bouden, T., & Ramdani, M. (2020). Face–iris multimodal biometric identification system. Electronics9(1), 85.

Tyagi, S., Chawla, B., Jain, R., & Srivastava, S. (2022). Multimodal biometric system using deep learning based on face and finger vein fusion. Journal of Intelligent & Fuzzy Systems42(2), 943-955.

- From the problem gap, you need to define the aim and objectives and then show how the chosen techniques, CNN and other feature extraction techniques will be used to achieve this aim.

- The contribution of the papers needs to be clearly stated in the introduction too. 

- The presented literature review is a very brief summary of the related papers. You need to show the main limitations/problems of these papers. You may find a common problem among all of them. Then you need to write one paragraph at the end to show the main problem gap which your paper will address (why there is still a need to propose a multi-modal of face and finger veins? what your proposal will address/improve?)

- The proposed model is not clear. Why there is not face extraction/segmenation like the veins? what are the pre-processing techniques used? what are the feature extraction techniques used? why are those techniques? Figure 1 is not a good representative of the model. 

- In section 3, you are still talking about why the face will be combined with the veins feature, this should be covered in the problem description. This section should be talking about the proposed system/model and showed clearly describe how it works. 

- The images' quality is not good.

- It is not clear which performance metrics were used. The authors mentioned accuracy, precision, recall, Sensitivity, Specificity, and ROC but there are no results of these metrics. Also, why precision/Specificity while they are doing the same thing. Why and (recall and Sensitivity).

- Also, the results are not well-structured to measure the efficiency of the system. Not clear, it is expected to see a comparison between unimodal identification and multi-modal identification as well as compassion with related work (I gave examples above).

- The conclusion is too short and does not show all need information.

- The English language needs careful proofreading. For example,

- in the abstract the following sentence is not correct/clear: "The weights of the two biological features were obtained through the self-attention mechanism, which combined with the ResNet residual structure combined the self-attention weight feature Channel concat cascade with the bimodal fusion feature".

- leave a space between the citation and the last word, e.g., attention to information security[1].

- whenever you give information/fact you need to put a reference, e.g., "With the upgrading and development of biometric identification technology, biometric identification is constantly changing from single to multi-modal, and multi-modal biometric identification technology has become a market development trend". You need support this information with a reference. 

Author Response

Thank you very much for your valuable comments on the paper.

The reply comments are in the attachment.

Thank you again for your comments.

Reviewer 2 Report

This paper presents an experimental evaluation of a multimodal biometric system, fusioning face and finger veins biometric data, compared against the same biometric data taken seprately, done over two well-known variants of Convolutional Neural Networks.

Although the proposal has some merit, the presentation is not good enough to fully appreciate it. On one hand, the English writing is rather poor: there are even some paragraphs that are unintelligible, making a review by a native English speaker necessary. Also, there are many minor mistakes and inconsistencies trough out the document, some of which are indicated in the attached PDF file.

However, the most relevant issues are the following. On one hand, the experimental setting (or, at the very least its discussion) is very poor: apparently the experiments are run ONLY ONCE, which of course give insufficient support to draw any conclusion from its results. On the other hand, there are several claims with insufficient support that, even when they perhaps could be drawn from the results, are not explained and discussed properly to reach them. Finally, the contribution of the paper is not discussed clearly enough. While it is understandable that the fusion of two biometrics could improve the performance of a biometric system against the use of such features by separate, this is not a new idea. Also, the use of deep learning for image classification in general, and the use of the AlexNet and VGG-19 architectures of CNNs in particular, is widely adopted. Thus, what this paper adds to the state of the art is not sufficiently clear.

I would strongly suggest the authors that they continue to improve their proposal for it to be of the adequate quality for publication.

Author Response

Thank you very much for your suggestions.

The paper has been revised.

I have replied to the documents you sent one by one, which are in the attachment.

Thank you again for your suggestions.

Reviewer 3 Report

The introduction section could provide more detailed background information.

On page 2, please check the numbering of the references. For example, after Ref [4], in line 67, why there is Ref [22]?

Could you elaborate on how your approach outweighs other similar approaches in the existing literature?

The conclusion section could have been expanded.

The references’ formatting is not following the same style and needs to be revised.

The paper requires proofreading.

Author Response

Thank you very much for your suggestions.

Thank you again for your suggestions.

Reviewer 4 Report

The authors present a novel method for fusion of finger vain and face authentication at the feature level. The article is generally good but there are some minor issues that should be addressed before publication:

1) Check the spelling and grammar. While the article is written well in general there are some minor mistakes, repetions and grammatically wrong sentences.

2) The ROC plots showing the results should be improved. Figures 7,8 and 9 look very similar. Maybe the main differences lie in the initial part of the graph. The initial section of the graphs should be magnified and the difference between results better underlined. Specifically when commenting the figures, the authors should emphasise the advantage of the fusion approach compared to the unimodal approach. It is not evident from the plots.

3) The conclusion section is too short. Expand the conclusions to emphasise the main contribution of the paper and the amount of improvement offered by the proposed method. Also some future research plans should be added.

Author Response

Thank you very much for your valuable suggestions.

Round 2

Reviewer 2 Report

The paper has improved from the previous version, yet is still far from ready for publication.

Many of the English writing has been revised, however there remain many small errors and several paragraphs that are hard to understand. Again, I very STRONGLY recommend a revision by a native English speaker.

The central issue of this paper remains to be addressed: the contribution is not sufficiently stressed, making it difficult to see why this work should be published. As stated in the previous revision, the idea of fusing several biometrics into one single system in order to improve its performance is not new: the authors even cite a work in their Related Work claiming this. In particular, some of the works cited fuse face and finger vein data together or with other biometric data, making this not a novel idea. Also, the use of deep learning for image classification in general, and the use of the AlexNet and VGG-19 architectures of CNNs in particular, is widely adopted. Thus, what this paper adds to the state of the art remains unclear.

Additionally, several choices for the experimental setting and results analysis (in particular the experimental validation method and the performance metrics) are not explained sufficently, giving little support for their choosing. Some more detailed comments appear in the attached PDF file.

Again, I would strongly suggest the authors that they continue to improve their proposal for it to be of the adequate quality for publication.

Author Response

Thank you very much for your valuable comments.

Your comments have been answered in the attachment.
